# FreqMark: Invisible Image Watermarking via Frequency Based Optimization in Latent Space

**Yiyang Guo**[*1,5†], **Ruizhe Li**[*2], **Mude Hui**[3], **Hanzhong Guo**[4], **Chen Zhang**[1]
**Chuangjian Cai**[5], **Le Wan**[5], **Shangfei Wang**[‡1]

[1]University of Science and Technology of China [2]Fudan University
[3]University of California, Santa Cruz [4]The University of Hong Kong [5]IEG, Tencent
{guoyiyang, zhangchenzc}@mail.ustc.edu.cn; lirz22@m.fudan.edu.cn;
muhui@ucsc.edu; hanzhong@connect.hku.hk;
{herbertcai, vinowan}@tencent.com;
sfwang@ustc.edu.cn

## Abstract

Invisible watermarking is essential for safeguarding digital content, enabling copyright protection and content authentication. However, existing watermarking methods fall short in robustness against regeneration attacks. In this paper, we propose a novel method called FreqMark that involves unconstrained optimization of the image latent frequency space obtained after VAE encoding. Specifically, FreqMark embeds the watermark by optimizing the latent frequency space of the images and then extracts the watermark through a pre-trained image encoder. This optimization allows a flexible trade-off between image quality with watermark robustness and effectively resists regeneration attacks. Experimental results demonstrate that FreqMark offers significant advantages in image quality and robustness, permits flexible selection of the encoding bit number, and achieves a bit accuracy exceeding 90% when encoding a 48-bit hidden message under various attack scenarios.

## 1 Introduction

As the development of generative models [42, 32], distinguishing between AI-generated and real images becomes increasingly challenging, which brings new risks such as deepfakes and copyright infringement [5, 31]. By adding invisible watermarks within images, the concealed message can be tracked and used for purposes such as copyright verification, identity authentication, copy control, etc., thereby safeguarding against the misuse of image content.

Traditional methods [12, 15, 14, 36, 15] conceal hidden messages in the frequency space of images, providing resistance to Gaussian noise attacks but susceptibility to brightness, contrast, and regeneration attacks. The rapid development of deep learning has also promoted the iteration of watermarking techniques with enhanced robustness to numerous attacks. A common approach is to train a watermark embedding network through supervised learning to introduce imperceptible perturbations to images, and then to retrieve the hidden messages through a decoding network [46, 10, 55, 3]. An alternative method sets the optimization objective as the image itself, utilizing pre-trained neural networks to calculate the perturbations to be added, thereby bypassing the higher cost of training networks and providing a more flexible approach [29, 20]. However, with the continued advancement of generative models, leveraging their generalization capabilities to denoise watermarked images through regeneration attacks has proven to be an effective method for watermark removal [58, 43]. A logical step is to encode the watermark in the image latent space, allowing the watermark to

---

[*]Equal Contribution. [†]Work is done during an internship at IEG, Tencent. [‡]Corresponding Author.

38th Conference on Neural Information Processing Systems (NeurIPS 2024).

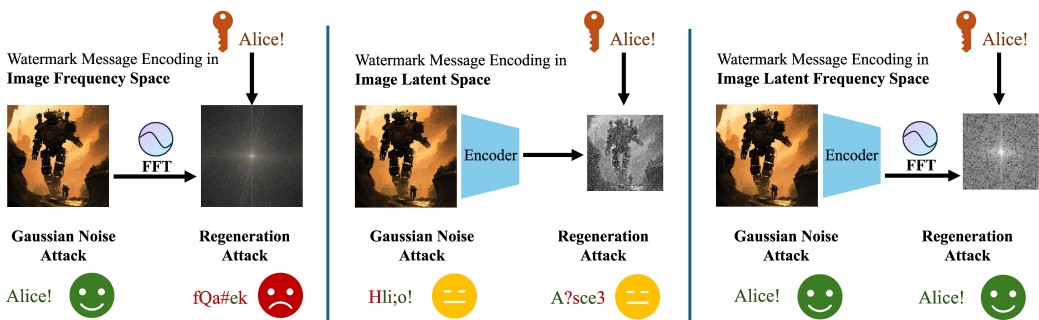

Figure 1: The robustness of different watermark encoding positions. **Left**: Encoding in image frequency space resists Gaussian noise but is vulnerable to regeneration attacks. **Middle**: Encoding in image latent space enhances resistance to regeneration attacks but introduces vulnerabilities to Gaussian noise. **Right**: FreqMark encodes latent frequency space in the image, achieving a strong defense against regeneration and traditional attacks.

be incorporated into the latent semantics of images. This approach improves robustness against regeneration attacks but increases susceptibility to Gaussian noise.

In this paper, we propose FreqMark, a novel self-supervised watermarking approach that amalgamates the benefits of frequency domain space and latent space, endowing it with a dual-domain advantage. Specifically, it utilizes fixed pre-trained Variational Autoencoder (VAE) [28] to embed watermark messages into images by making subtle adjustments in the latent frequency space. These messages are then decoded using a fixed pre-trained image encoder. Figure 1 illustrates the primary motivation behind our method. Introducing perturbations to watermark images in the latent and frequency domains offers distinct advantages. By optimizing the latent frequency domain of the image, we combine both approaches to effectively leverage their strengths, achieving a synergistic effect where the whole is greater than the sum of its parts.

We evaluate the performance of FreqMark on the DiffusionDB [49] and ImageNet [16] datasets, experimental results demonstrated that FreqMark achieves strong robustness while maintaining image quality, configurable payload capacity, and flexibility. With a 48-bit encoding setting, the bit accuracy can exceed 90% under various attacks. This performance indicates significant advantages over baseline methods, particularly excelling during regeneration attacks [7, 13, 58].

**Contributions:** (1) We propose a novel invisible image watermarking method named FreqMark, which encodes hidden messages within the latent frequency space of images. FreqMark achieves watermark embedding through indirect optimization centered on the image itself without requiring network training. (2) FreqMark is highly flexible, allowing for a free trade-off between the bits number of the encoded message, image quality and watermark robustness to meet diverse requirements. (3) FreqMark demonstrates significant robustness advantages, particularly during regeneration attacks compared to baseline methods. Experimental results validate the superiority of our proposed method.

## 2 Related Work

**Generative Models**   For a long time, Generative Adversarial Networks (GANs) [27, 1, 26, 22, 41, 33] have dominated image generation. Recently, diffusion models have emerged as a solid alternative to GANs for image generation [25, 17, 34, 35]. These models show significant improvements and applications across various domains. DDIM sampling [45] and latent diffusion [42] further accelerate the generation progress, while ControlNet [56] provides a powerful, controllable generation method. As high-quality image generation becomes more accessible, digital watermarking gains importance for protecting intellectual property rights and ensuring content authenticity.

**Image Watermarking**   The research history of image watermarking techniques is extensive. Early methods employ hand-crafted methods to hide messages within the spatial or frequency domain of images. Frequency-domain-based techniques typically exhibit better robustness and have been widely applied even before the rise of deep learning [12, 15, 14, 36, 15, 4]. The widely used open-source

model, Stable Diffusion [42], employs DwtDctSvd [14] as its default watermarking method. However, this method has been demonstrated to be relatively vulnerable to various attacks [58].

With the development of deep learning, methods utilizing encoder-decoder architectures have gained prominence. HiDDeN [60] and RivaGAN [55] train encoders to embed watermark messages into images and decoders to extract them, thereby enhancing robustness against noise while preserving image quality. RedMark [3] and StegaStamp [46] improve robustness by integrating a series of differentiable perturbations. RoSteALS [10] enhances the robustness of the watermark by fine-tuning the secret encoder and secret decoder in the latent space. In recent years, several innovative methods have been introduced. WatermarkDM [59] trains a diffusion model on watermarked images to create detectable watermarked images. Stable Signature [19] fine-tunes the decoder of a latent diffusion model to embed specific hidden messages. Tree-Ring [50] adopts a unique approach by encoding a particular pattern shape in the initial noise frequency space during the diffusion process and utilizes DDIM inversion [45] to detect watermarks. Differently, FNNS [29] and SSL [20] achieve message encoding by optimizing the image itself. However, the aforementioned methods struggle to strike a perfect balance between flexibility and robustness. In contrast, FreqMark offers higher degrees of freedom, allowing for a better trade-off among task requirements, and demonstrates robustness against regeneration attacks.

## 3   Background

FreqMark operates on the following scenario: A user embeds a $k$-bit watermark message $m_e$ into an image $I$ to get the watermarked image $I_w$ for which they hold the copyright. Upon discovering unauthorized usage of the image $I_w$, the user decodes the infringed image to obtain the message $m_d$, which serves as evidence to prove their ownership of the image's copyright.

Assuming that each bit of the decoded message from clean images is independent and has an equal probability of being -1 or 1, this method allows us to mathematically calculate the False Positive Rate (FPR) of decoding.

Let the encoded message be $m_e \in \{-1, 1\}^k$ and the decoded message be $m_d \in \{-1, 1\}^k$. The function $M(m_e, m_d)$ measures the number of matching bits between $m_e$ and $m_d$. Given the assumption regarding the image encoder output mentioned earlier, each bit of the decoded message from clean images is independent and follows Bernoulli random variables with a probability of 0.5 [19]. By set a decoding threshold $\tau \in \{0, \ldots, k\}$, once $M(m_e, m_d) \geq \tau$, we consider that the image has encoded the message $m_e$. Consequently, $M(m_e, m_d)$ follows a binomial distribution $B(k, 0.5)$. The final message will be transformed to $\{0, 1\}^k$ by applying function $f(x) = (x + 1)/2$ for easier processing of the binary format.

We could test the hypothesis $H_1$: *the image $x$ has hidden watermark*, and against the null hypothesis $H_0$: *the image $x$ has no hidden watermark*. From this, we can obtain a closed-form solution for the FPR under the threshold $\epsilon(\tau)$ using the regularized incomplete beta function $I_x(a; b)$:

$$\epsilon(\tau) = P(M(m_e, m_d) > \tau | H_0) = \frac{1}{2^k} \sum_{i=\tau+1}^{k} C_k^i = I_{1/2}(\tau + 1, k - \tau). \tag{1}$$

Based on the above formula, FPR $\approx 1.65 \times 10^{-6}$ when $k = 48$, $\tau = 39$; FPR $\approx 5.04 \times 10^{-8}$ when $k = 48$, $\tau = 41$. The results of watermark detection True Positive Rate (TPR) under various FPR settings are shown in Figure 8 of Appendix A.3.

## 4   Method

### 4.1   Overview

Figure 2 presents an overview of FreqMark. FreqMark employs a strategic methodology to embed invisible watermarks in images by adding perturbations in the latent frequency space. Specifically, we utilize a Variational Autoencoder (VAE) [28] to encode images into representations and then transform these image latents into the frequency domain. Only the perturbation within the latent frequency space of images is trained during watermark encoding and all the networks are fixed. A

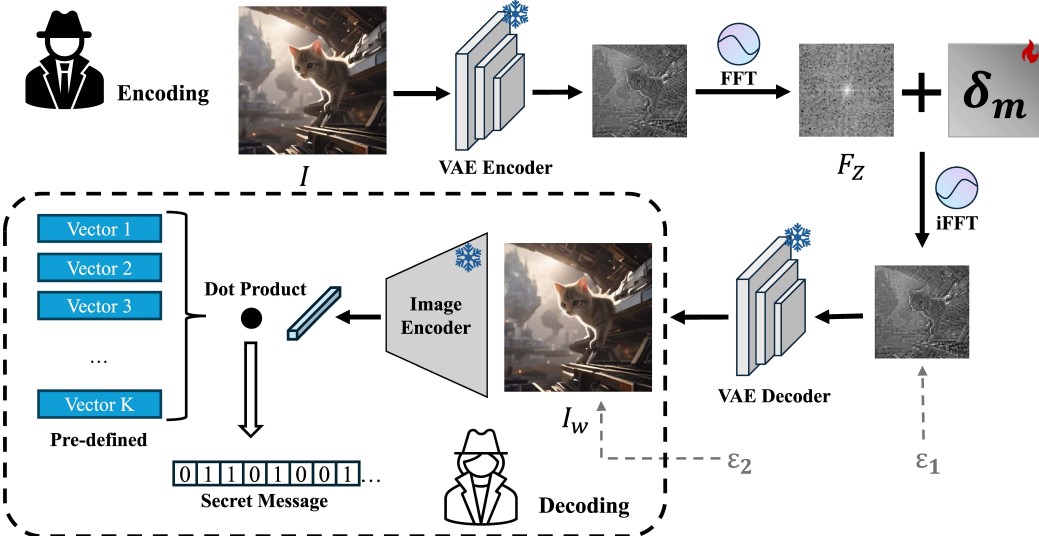

Figure 2: Overview of FreqMark. **Encoding**: FreqMark employs a pre-trained VAE model to encode watermarks within the latent frequency space of the image. $\epsilon 1$ and $\epsilon 2$ are Gaussian noise perturbations added during training. All networks are fixed and only perturbation $\delta_m$ is trained. **Decoding**: FreqMark utilizes a pre-trained image encoder to extract features from the image and extracts the watermark by comparing this feature against predefined directions.

pre-trained image encoder is used for watermark extraction, the features of the watermarked image obtained from the encoder are compared with predefined directional vectors to reveal the hidden message.

In the optimization process, the image quality is maintained with minimal degradation by utilizing PSNR and LPIPS loss [57], while the watermark message is constrained by using hinge loss. In addition, noises and augmentation are introduced for enhanced robustness. Figure 3 shows some watermark image examples from FreqMark.

## 4.2 Message Embed in Latent Frequency Space

Some current methods embed hidden messages by directly optimizing image pixels [20, 29], which leads to poor robustness against regeneration attacks [58]. FreqMark involves strategically adding perturbations in the frequency domain of the image latent representation to embed an invisible watermark. These perturbations in the frequency domain are more concealed than those in the pixel space, making them harder to eliminate and thus offering superior robustness against regeneration attacks while minimizing the impact on image quality.

First, we use a pre-trained VAE encoder $E$ to encode the image into a latent image and then transform the latent image into the frequency domain using Fast Fourier Transform (FFT), represented as:

$$F_Z = FFT(E(I)), \tag{2}$$

where $I$ is the input image, $E(I)$ is the latent representation encoded by the VAE encoder $E$, and $F_Z$ is the frequency domain latent representation of the image.

Next, we embed the hidden message into the frequency domain latent representation $F_Z$ by adding a slight perturbation, then decode the frequency domain latent representation back into the image using inverse Fast Fourier Transform (iFFT) $FFT^{-1}$ and the VAE decoder $D$, as follows:

$$I_w = D(FFT^{-1}(F_Z + \delta_m)), \tag{3}$$

where $I_w$ is the watermarked image, $\delta_m$ is the perturbation added in the frequency domain of the latent image for embedding the watermark.

### 4.3 Decoding by Pre-trained Image Encoder

During the decoding process, similar to SSL[20], we predefine a set of K N-dimensional vectors $V_K^N = \{v_1, v_2, ..., v_K \mid K \leq N\}$ within the feature space of the pre-trained image encoder $E_{img}$. For images with embedded watermarks $I_w$, the feature vector $z_{I_w} = E_{img}(I_w)$ is obtained after passing through $E_{img}$. By calculating the signs of the dot product between the direction vectors $V_K^N$ and $z_{I_w}$, we can extract the hidden message $m_d$:

$$m_d^k = sign(z_{I_w} \cdot v_k) = sign(E_{img}(I_w) \cdot v_k), v_k \in V_K^N, \tag{4}$$

where $m_d^k$ represents the $k$-th bit of $m_d$ and $v_k$ denotes the $k$-th direction vector of $V_K^N$, $sign(x) = 1$ when $x >= 0$; $sign(x) = -1$ when $x < 0$.

One significant advantage of this decoding method is its flexibility in setting the number of watermark bits. The number of direction vectors determines the encoding bits. Using the technique mentioned above of optimizing images within the frequency domain, we can ensure the robustness of the hidden messages while significantly minimizing the quality impact on the image.

### 4.4 Training Objective

Our goal is to embed an invisible watermark by optimizing the frequency map of the image latent $F_Z$, where the perturbation $\delta_m$ serves as the trainable parameter, with all pre-trained networks remaining fixed. The optimization via perturbations ensures both watermark robustness and the preservation of image quality, offering flexibility in encoding bits and embedding strength.

**Image Quality**  We utilize PSNR loss to reduce discrepancies between the watermarked image and the original image and also incorporate LPIPS loss [57] to make alterations less perceptible.

$$\mathcal{L}_p = -PSNR(I_w, I), \tag{5}$$

$$\mathcal{L}_i = LPIPS(I_w, I). \tag{6}$$

**Watermark Message**  The optimization goal is to modify the image features $z_{I_w}$ processed by the pre-trained image encoder $E_{img}$, aligning them on the K direction vectors $V_K^N$ to correspond with the encoded message. We define message loss as the hinge loss with margin $\mu \geq 0$ on the projections:

$$\mathcal{L}_m(I_w) = \frac{1}{K} \sum_{k=1}^{K} max(0, (\mu - (z_{I_w} \cdot v_k) \cdot m_k)), v_k \in V_K^N, \ m_k \in \{-1, 1\}. \tag{7}$$

### 4.5 Robustness Enhancement Strategy

During the training process, we employ an augmentation strategy by adding Gaussian noise with a mean of 0 and standard deviations of $s1$ and $s2$ to the latent space and pixel space, represented as $\epsilon 1$ and $\epsilon 2$, respectively. The result is the acquisition of the perturbed images $I_{p1}$ and $I_{p2}$:

$$I_{p1} = D(FFT^{-1}(F_Z + \delta_m) + \epsilon 1), \tag{8}$$

$$I_{p2} = D(FFT^{-1}(F_Z + \delta_m)) + \epsilon 2. \tag{9}$$

The final loss is defined as:

$$\mathcal{L} = \mathcal{L}_m(I_w) + \mathcal{L}_m(I_{p1}) + \mathcal{L}_m(I_{p2}) + \lambda_p \mathcal{L}_p(I_w, I) + \lambda_i \mathcal{L}_i(I_w, I), \tag{10}$$

where $\lambda_p$ and $\lambda_i$ are the respective weights for each loss function.

By incorporating corresponding attacks during training, we optimize the solution space for hidden message embedding, enabling the image to withstand a broader range of attacks. Experimental results reveal that this straightforward approach can effectively enhance the model's robustness against regeneration attacks.

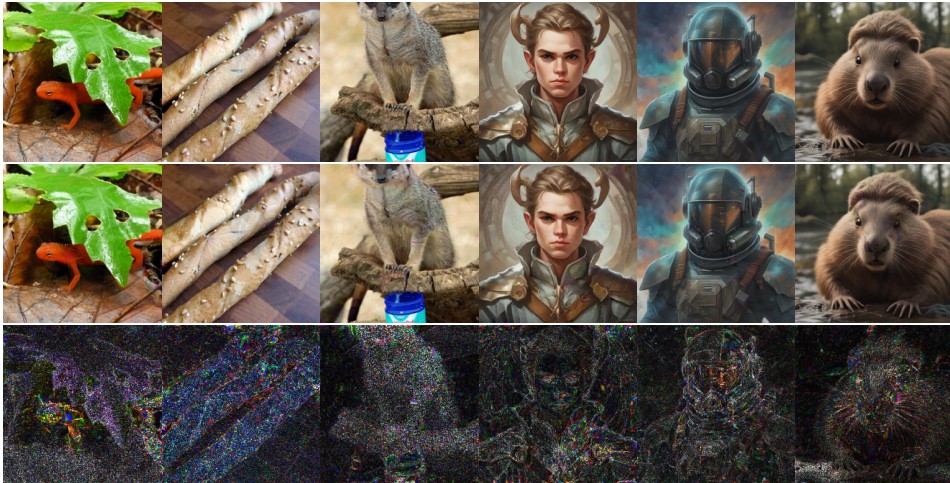

Figure 3: Examples of watermarked images. The first three columns are from ImageNet [16], and the others are generated by the prompts from DiffusionDB [49]. These watermarked images have 48-bit messages and are robust to various attacks. **Top**: origin image. **Middle**: watermarked image. **Bottom**: pixel difference (amplified by a factor of 10 to enhance the visual effect).

## 5  Experiments

In this section, we evaluate our method based on the image quality and robustness metrics under various attacks, comparing it with the baseline methods. Additionally, ablation studies and analyses are carried out to explore the process deeply. In our experiments, KL auto-encoder [28] from Stable Diffusion [42] is used as the pre-trained VAE, and DINO v2 small [37] is used as the pre-trained image encoder. More implementation details are in the Appendix A.1.

### 5.1  Experimental Setup

**Datasets**   A test dataset is compiled, consisting of 500 images randomly selected from the ImageNet validation set [16], in conjunction with 500 images generated using Stable Diffusion [42] based on prompts from the DiffusionDB [49] dataset. This diverse data collection enables a thorough assessment of the performance across various scenarios and perspectives.

**Comparison Methods**   Three methodologies are selected for comparison. DwtDctSvd [14] is a classic frequency-domain-based approach and the default watermarking method in Stable Diffusion [42]. Stable Signature [19] embeds specific hidden messages by fine-tuning the VAE decoder of Stable Diffusion, enabling the watermarking process to be integrated with the generation process. SSL [20] is the latest method that employs image pixel optimization to embed watermarks. We employ their default configurations, wherein the PSNR threshold of SSL is set to 31 dB for a convenient comparison of bit accuracy.

**Evaluation Metrics**   Following previous work [19, 20, 58], PSNR and SSIM [48] are used as the image quality benchmark. To evaluate robustness, we utilize bit accuracy as a metric to measure the degradation of hidden messages under diverse attacks. Various attack methods have been implemented, including a brightness change of 0.5, a contrast change of 0.5, 50% JPEG compression, Gaussian blur with a kernel size of 5, and Gaussian noise with $\sigma = 0.05$. Moreover, the experiments incorporate two types of VAE regeneration attacks [7, 13] from the CompressAI library [9] with a compression factor of 3, and a diffusion regeneration attack is carried out with 60 denoising steps [58].

### 5.2  Benchmarking Watermark Accuracy and Image Quality

Table 1 displays the image quality and bit accuracy following watermark embedding on ImageNet [16] and DiffusionDB [49] datasets. The classic method DwtDctSvd [14] shows robustness against Gaussian attacks but poor performance against others, while the others show better robustness against

Table 1: Performance of different watermarking methods on ImageNet and DiffusionDB.

| Method | PSNR | SSIM | Bit Accuracy | | | | | | | | | |
| --- | --- | --- | --- | --- | --- | --- | --- | --- | --- | --- | --- | --- |
| | | | None | Brightness | Contrast | JPEG | Gau. blur | Gau. noise | VAE-B | VAE-C | Diffusion | Avg |
| ImageNet | | | | | | | | | | | | |
| DwtDctSvd[14] | **39.67** | **0.978** | 0.993 | 0.636 | 0.489 | 0.848 | 0.992 | **0.993** | 0.550 | 0.562 | 0.592 | 0.739 |
| ±std | 1.939 | 0.011 | 0.049 | 0.307 | 0.222 | 0.147 | 0.058 | 0.051 | 0.063 | 0.078 | 0.106 | N/A |
| SSL Watermark[20] | 31.04 | 0.862 | **1.000** | **1.000** | **1.000** | 0.972 | **1.000** | 0.937 | 0.793 | 0.777 | 0.743 | 0.914 |
| ±std | 0.110 | 0.029 | 0.000 | 0.000 | 0.000 | 0.034 | 0.000 | 0.028 | 0.073 | 0.096 | 0.077 | N/A |
| Stable Signature[19] | 28.74 | 0.838 | 0.978 | 0.971 | 0.937 | 0.832 | 0.859 | 0.892 | 0.630 | 0.645 | 0.534 | 0.809 |
| ±std | 3.246 | 0.080 | 0.054 | 0.061 | 0.092 | 0.106 | 0.121 | 0.117 | 0.086 | 0.105 | 0.064 | N/A |
| FreqMark(Ours) | 31.27 | 0.857 | **1.000** | 0.995 | **1.000** | 0.991 | **1.000** | 0.939 | **0.938** | **0.924** | **0.969** | **0.973** |
| ±std | 3.359 | 0.038 | 0.000 | 0.028 | 0.000 | 0.024 | 0.000 | 0.088 | 0.083 | 0.081 | 0.052 | N/A |
| DiffusionDB | | | | | | | | | | | | |
| DwtDctSvd[14] | **39.49** | **0.978** | 1.000 | 0.607 | 0.457 | 0.887 | **1.000** | **1.000** | 0.563 | 0.556 | 0.569 | 0.738 |
| ±std | 1.182 | 0.006 | 0.000 | 0.308 | 0.194 | 0.109 | 0.000 | 0.000 | 0.053 | 0.059 | 0.085 | N/A |
| SSL Watermark[20] | 31.01 | 0.827 | 1.000 | **1.000** | **1.000** | 0.956 | **1.000** | 0.954 | 0.742 | 0.744 | 0.729 | 0.903 |
| ±std | 0.064 | 0.027 | 0.000 | 0.000 | 0.000 | 0.048 | 0.000 | 0.037 | 0.109 | 0.102 | 0.081 | N/A |
| Stable Signature[19] | 28.31 | 0.844 | 0.996 | 0.996 | 0.990 | 0.896 | 0.858 | 0.967 | 0.668 | 0.733 | 0.527 | 0.848 |
| ±std | 1.608 | 0.033 | 0.013 | 0.012 | 0.014 | 0.042 | 0.086 | 0.028 | 0.063 | 0.049 | 0.040 | N/A |
| FreqMark(Ours) | 31.20 | 0.854 | **1.000** | **1.000** | **1.000** | **1.000** | **1.000** | 0.934 | **0.925** | **0.897** | **0.945** | **0.967** |
| ±std | 1.538 | 0.029 | 0.000 | 0.000 | 0.000 | 0.000 | 0.000 | 0.061 | 0.066 | 0.059 | 0.047 | N/A |

Table 2: Comparison of image quality between VAE and FreqMark.

| Dataset | PSNR | SSIM |
| --- | --- | --- |
| VAE | | |
| ImageNet [16] | $31.37 \pm 4.59$ | $0.868 \pm 0.085$ |
| DiffusionDB [49] | $31.22 \pm 1.96$ | $0.879 \pm 0.032$ |
| FreqMark | | |
| ImageNet [16] | $31.27 \pm 3.36$ | $0.857 \pm 0.038$ |
| DiffusionDB [49] | $31.20 \pm 1.54$ | $0.854 \pm 0.029$ |

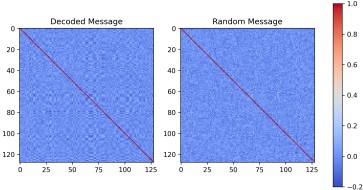

Figure 4: The correlation matrix of each bit of the decoded message from the vanilla images and the random message.

brightness, contrast, and JPEG attacks but still poor in regeneration attacks. FreqMark demonstrates exceptional robustness against regeneration attacks with acceptable image quality.

We also calculate the PSNR and SSIM of the images from two datasets after VAE reconstruction. The data in Table 2 demonstrates that the impact of FreqMark on image quality is limited. Moreover, the standard deviation of image quality after FreqMark processing is also reduced to some extent.

In order to verify the hypothesis proposed in Section 3, we randomly select 2,000 clean images to perform message decoding in 128-bit and calculate the correlation coefficients between each bit, comparing them to random messages. Figure 4 shows that different bits with near-zero correlation coefficients can be considered as independent random variables.

Notably, FreqMark allows users to adjust image quality and encoding bits, showing remarkable robustness while maintaining acceptable image quality, particularly against VAE and diffusion regeneration attacks [9, 7, 13, 58]. We will further evaluate the robustness of FreqMark on the DiffusionDB dataset [49] in Section 5.4. The ablation studies in Section 5.5 will explore the impact of parameter adjustments on robustness.

## 5.3 Why the Frequency Domain of Image Latent Space

To further illustrate the advantages of optimizing in the frequency domain of the image latent space, We conducted experiments using the same settings to optimize the pixel, the image latent space, and the frequency domain of image pixels in the DiffusionDB dataset [49]. Simultaneously, enhancements in pixel and latent domains are retained (if the method involves the latent space). A consistent average image quality range is maintained for comparison purposes.

As illustrated in Figure 5 and Table 3, optimizing the frequency domain of image pixels provides the watermarked image with robustness against all attacks except the regeneration attack. In contrast,

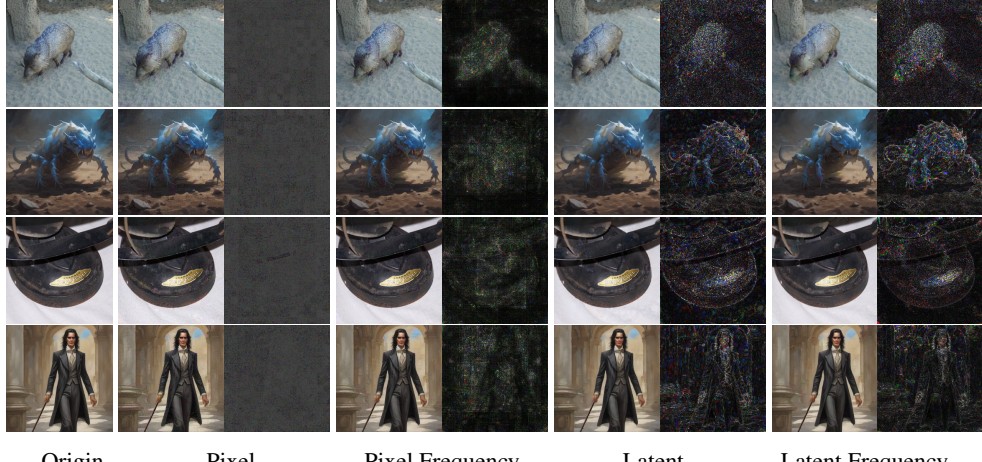

| Origin | Pixel | Pixel Frequency | Latent | Latent Frequency |

Figure 5: Watermarked images under different optimization locations. We compared four distinct optimization objectives for watermark embedding, including the image pixel domain, the frequency domain of the image pixel, the image latent space, and the frequency domain of the image latent space (ours). The difference after watermarking addition is amplified by a factor of 10.

Table 3: Performance of different optimization locations.

| Location | PSNR | SSIM | Bit Accuracy | | | | | | | | | |
| | | | None | Brightness | Contrast | JPEG | Gau. blur | Gau. noise | VAE-B | VAE-C | Diffusion | Avg |
|---|---|---|---|---|---|---|---|---|---|---|---|---|
| Pixel | **31.36** | 0.771 | 0.950 | 0.935 | 0.937 | 0.848 | 0.885 | 0.925 | 0.642 | 0.654 | 0.542 | 0.813 |
| Pixel Frequency | 31.31 | 0.809 | **1.000** | **1.000** | **1.000** | 0.950 | 0.937 | **1.000** | 0.797 | 0.775 | 0.596 | 0.895 |
| Latent | 31.35 | **0.886** | 0.994 | 0.993 | 0.981 | 0.906 | 0.979 | 0.804 | 0.796 | 0.833 | 0.675 | 0.885 |
| Latent Frequency | 31.20 | 0.854 | **1.000** | **1.000** | **1.000** | **1.000** | **1.000** | 0.934 | **0.925** | **0.897** | **0.945** | **0.967** |

images obtained by optimizing the image latent space appear more natural and smooth, with a strong correlation to the semantic information of the image. This approach demonstrates robust potential against diffusion attacks, indicating that both optimization methods have unique features. Combining the advantages of both approaches, FreqMark optimizes the frequency space of the image latent. The watermark is closely related to local patterns while retaining the characteristics of frequency domain optimization. Combining these two aspects produces a synergistic effect, making FreqMark robust against regeneration attacks.

### 5.4 Further Robustness Results

**Diffusion Attack Steps** We evaluate the impact of diffusion attacks of varying intensities on bit accuracy and also calculate the PSNR between the attacked image and the original watermarked image. Table 4 indicates that FreqMark maintains commendable robustness even under higher intensity attacks.

Table 4: Performance under Different Diffusion Steps.

| Diffusion Steps | 60 | 80 | 100 | 120 | 140 | 160 | 180 | 200 |
|---|---|---|---|---|---|---|---|---|
| Bit Acc | 0.945 | 0.863 | 0.831 | 0.754 | 0.712 | 0.692 | 0.660 | 0.637 |
| PSNR | 27.67 | 26.95 | 26.19 | 25.46 | 24.92 | 24.47 | 23.99 | 23.57 |

**Vae Attack Strength** We employ the same VAE used in the watermarking process of FreqMark to conduct perturbation attacks. Table 5 demonstrates that FreqMark exhibits exceptional robustness under such attacks. This is because the perturbation on the latent FFT changes the overall distribution of the image latent, making the watermark message affect the entire image globally. Therefore, it

Table 5: Performance under VAE Attack in Latent FFT Domain and Gaussian Noise Disruption in Pixel FFT Domain.

| | VAE Attack (Latent FFT) | | | | | Gaussian Noise Attack (Pixel FFT) | | | | |
|---|---|---|---|---|---|---|---|---|---|---|
| PSNR | 31.43 | 30.31 | 28.98 | 27.39 | 25.82 | 31.09 | 29.68 | 28.04 | 26.46 | 25.04 |
| Bit Acc | 1.000 | 1.000 | 0.998 | 0.990 | 0.975 | 1.000 | 1.000 | 1.000 | 1.000 | 1.000 |

is difficult for perturbations on the latent of image to damage the watermark message. A similar phenomenon is observed in the pixel FFT when facing the Gaussian Noise attack.

**Adversarial Attack**   Following the settings in [5], we apply adversarial attacks targeting the latent representations of the watermarked images.

$$max_{I_{adv}}|E(I_{adv}) - E(I)\|_2, s.t.\|I_{adv} - I\|_\infty \leq \epsilon \tag{11}$$

Table 6: Performance under Different Adversarial Attack Strength.

| Attack Strength (eps) | Bit Acc | TPR@0.1%FPR |
|---|---|---|
| 2/255 | 1.000 | 1.000 |
| 4/255 | 0.987 | 1.000 |
| 6/255 | 0.944 | 0.986 |
| 8/255 | 0.893 | 0.972 |

Table 6 demonstrates that FreqMark exhibits strong robustness when facing adversarial attacks targeting latent representations. We believe that this can be attributed to the limited impact of attacks targeting latent representations on the latent FFT domain.

## 5.5   Ablation Studies

**Image Quality**   As watermarking always requires modifications to the image pixels, a trade-off between image quality and embedded message robustness is inevitable. FreqMark allows users to find a satisfactory balance. We control watermarked image quality within a specific range by adjusting the weight of the PSNR loss function. As shown in Figure 6a, FreqMark can achieve a PSNR that is nearly 2dB higher compared to the reconstruction obtained using VAE with robustness against varying attacks, maintaining strong performance with over 80% accuracy.

**Encoding Bits**   The number of bits in the encoding message significantly impacts robustness, with more bits embedded at a given image quality potentially reducing robustness. FreqMark allows users to adjust the watermark bit number based on their needs. Figure 6b shows bit accuracy for encoding bit numbers. We test robustness from 32 to 128 bits of watermark message in 16-bit increments, setting the PSNR of the watermarked image to about 31 dB. FreqMark also demonstrates strong robustness, with the lowest bit accuracy remaining above 0.75 even under regeneration attacks on images watermarked with 128-bit messages.

**Noising Scale**   We also study the impact of latent noise $\epsilon_1$ and pixel noise $\epsilon_2$ on robustness. Incorporating noise attacks during training can effectively improve robustness in Gaussian noise and regeneration attacks, but excessive noise may disrupt training and reduce performance. We analyze how noises in latent and pixel space affect robustness by fixing either $\epsilon_1$ or $\epsilon_2$ and changing the other standard deviation. As shown in Figure 6c and Figure 6d, the results confirm that introducing both types of noise positively influences robustness, and adding an appropriate amount of noise can significantly enhance resistance against regeneration attacks. We finally set $\epsilon_1$ to 0.25 and $\epsilon_2$ to 0.06 based on the experimental results. The complete experimental data can be found in Table 11 of the Appendix A.3.

**Spacial Perturbations**   The bit accuracy of FreqMark significantly declines under specific spatial transformation attacks, such as rotation and cropping. By incorporating relevant augmentations like

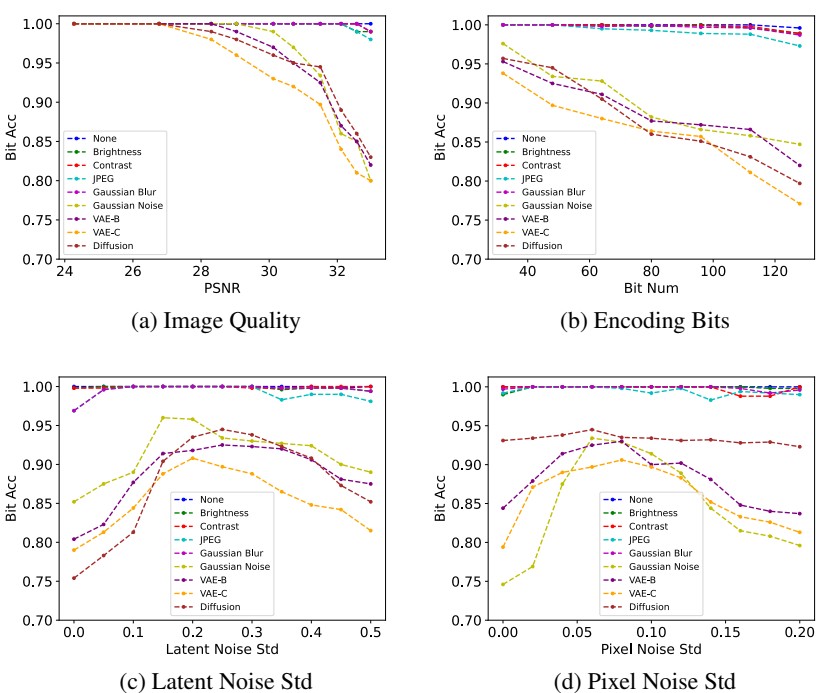

|  | (a) Image Quality | (b) Encoding Bits |
| :-: | :-: | :-: |
|  | (c) Latent Noise Std | (d) Pixel Noise Std |

Figure 6: Impact of different parameter settings on the robustness of the watermark. **(a)** Bit accuracy under different watermarked image quality (48 bits). **(b)** Bit accuracy under different encoding bits (31dB). **(c)** Bit accuracy under different latent noise $\epsilon_1$ ($\epsilon_2 = 0.06$). **(d)** Bit accuracy under different latent noise $\epsilon_2$ ($\epsilon_1 = 0.25$).

Table 7: Performance on Spatial Perturbations.

| Spatial Augmentations | Bit Acc | | |
| :-: | :-: | :-: | :-: |
|  | Resize 0.3 | Rotate 90 | Crop 0.7 |
| ✗ | 0.961 | 0.621 | 0.721 |
| ✓ | 0.968 | 0.927 | 0.921 |

random rotation and random cropping during training, we effectively bolster the robustness against these transformations, as evidenced in Table 7.

## 6 Conclusion

In this paper, we introduce a technique for embedding hidden messages in images by optimizing the frequency domain in the latent space, named FreqMark, providing remarkable robustness due to deeper perturbations. FreqMark allows user-defined encoding bits and watermark strength, striking an optimal balance between image quality and robustness.

**Limitations and Broader Impacts** FreqMark exhibits notable flexibility and robustness, yet there is room for optimization, particularly in the degree of naturalness in the fusion of watermarks and images. Future research can focus on improving image quality and human perceptibility constrained by the limitations of VAE, as a superior VAE would likely boost processing time and overall effectiveness. FreqMark presents an innovative image watermarking approach for image provenance, copyright protection, etc. However, like other image watermarking methods, it also necessitates the prevention of unauthorized misuse such as copyright abuse [5].

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

# A Appendix

## A.1 Implement details

**Hyperparameters** The KL auto-encoder from Stable Diffusion 2-1 [42] is utilized. Due to the significant reconstruction loss associated with low-resolution images, the images are upscaled to $512 \times 512$ for processing. In the watermark addition stage, the Adam optimizer is used with a learning rate of 2.0 and training for 400 steps. We set the PSNR loss weight $\lambda_p$ to 0.05 and the LPIPS loss weight $\lambda_i$ to 0.25. To encode the watermark, the first 128 dimensions of the output feature generated by the Dino v2 small image encoder [37] are utilized. In the experiments, we set the directional vectors as a set of 48 vectors, where the $i$-th vector has a value of 1 in its $i$-th dimension and 0 for the remaining dimensions. In addition, during the training phase, two types of spatial transformations and pixel noise are selected with equal probability. For rotation augmentation, the rotation angle is randomly chosen in 90-degree increments. The crop augmentation is set with a crop scale range of $[0.2, 1.0]$ and a crop ratio range of $[3/4, 4/3]$.

**Compute Resources** All experiments could be conducted on a single A-100 GPU with 40GB memory. Processing two images ($512 \times 512$) in parallel takes about 5 to 6 minutes (400 steps, fp32). Increasing the batch size will improve overall efficiency. Figure 7 illustrates the relationship between the number of training steps and performance. Experimental results indicate that FreqMark still exhibits satisfactory performance with fewer training steps.

**License** The assets and models used in this paper are all publicly available, including Stable Diffusion 2-1 (Open Rail++-M License) [42], DINO v2 (Apache-2.0 License) [37], DiffusionDB (CC0 1.0 License) [49], and ImageNet (Custom License, as viewed on https://www.image-net.org/download) [16].

## A.2 Additional Comparison Results

We additionally compare the performance of our method with two watermarking approaches different from FreqMark: StegaStamp [46], a classical encoder-decoder-based watermarking method, and TreeRing [50], a watermarking technique that combines watermark embedding with image generation without requiring any training. Specifically, we evaluate the results under similar image quality conditions (PSNR of 28) and the same encoded bit number (100 bits) for StegaStamp. Due to TreeRing is a 1-bit encoding method, the true positive rate at a 1% false positive rate (TPR@1%FPR) is specifically compared against TreeRing [50].

Table 8: Additional Comparison of StegaStamp [46] and Tree-Ring [50].

| Bit Acc | None | Bright | Contrast | JPEG | G.Blur | G.Noise | VAE-B | VAE-C | Diffusion | Avg |
|---|---|---|---|---|---|---|---|---|---|---|
| StegaStamp[46] | 0.999 | 0.999 | 0.998 | 0.994 | 0.997 | 0.991 | 0.981 | 0.984 | 0.857 | 0.978 |
| FreqMark(100 bits) | 1.000 | 1.000 | 0.999 | 0.999 | 0.998 | 0.989 | 0.976 | 0.934 | 0.933 | 0.981 |
| **TPR@1%FPR** | **None** | **Bright** | **Contrast** | **JPEG** | **G.Blur** | **G.Noise** | **VAE-B** | **VAE-C** | **Diffusion** | **Avg** |
| Tree-Ring[50] | 1.000 | 1.000 | 1.000 | 0.996 | 0.999 | 0.918 | 0.991 | 0.995 | 0.998 | 0.989 |
| FreqMark(48 bits) | 1.000 | 1.000 | 1.000 | 1.000 | 1.000 | 0.986 | 0.989 | 0.975 | 1.000 | 0.994 |

Compared to StegaStamp [46], FreqMark offers strong robustness against diffusion attacks and greater flexibility due to its network independence. Users can adjust image quality, encoding bits, and customize the decoding vectors to enhance the watermark security. As opposed to Tree-Ring [50], FreqMark is capable of encoding significantly more information (48 bits vs. 1 bit) at a similar TPR@1%FPR.

## A.3 Additional Experimental Results

**Training Steps** Figure 7 shows that FreqMark achieves considerable robustness even with fewer training steps, and there is no significant change in image quality throughout the process. Users can adjust the number of training steps according to their requirements as a trade-off.

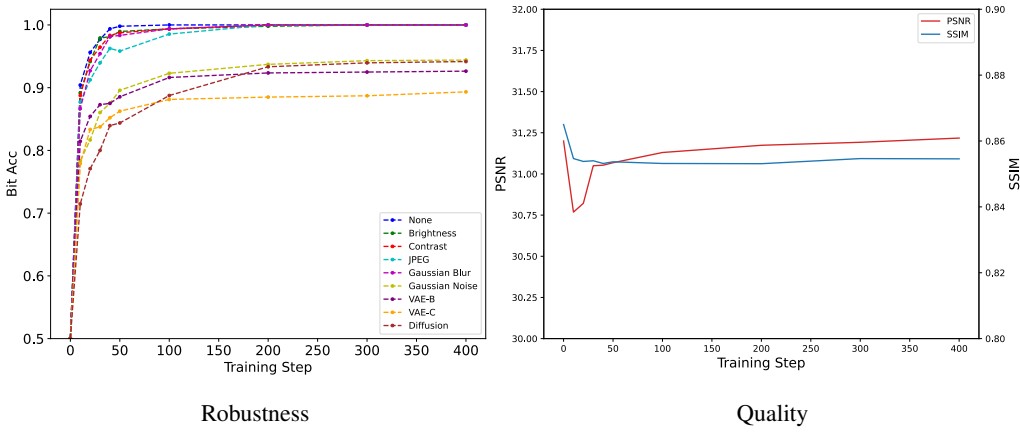

Robustness                                    Quality

Figure 7: The relationship between the training steps, watermark robustness, and image quality.

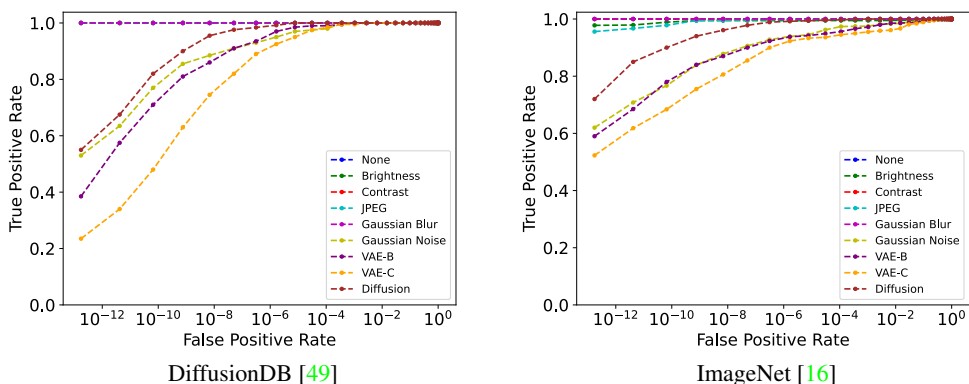

DiffusionDB [49]                              ImageNet [16]

Figure 8: The TPR/FPR curve under various attacks in two datasets.

**True Positive Rate vs. False Positive Rate**   We utilize 5,000 watermarked images to plot the FPR-TPR curve. Figure 8 shows that FreqMark exhibits remarkably high watermark detection accuracy in the range of FPR=$10^{-6}$ to $10^{-7}$. It is observed that the results for the DiffusionDB dataset [49] exhibit a significantly lower TPR at extremely low FPR compared to the ImageNet dataset [16]. This could be attributed to the higher standard deviation of bit accuracy in the ImageNet dataset [16], which consequently leads to a superior TPR under extreme conditions.

**Additional Quality Metrics**   We include CLIP-FID [39] and L2 as supplementary image quality metrics to further evaluate FreqMark's image quality on the DiffusionDB dataset [49]. Table 9 demonstrates that FreqMark exhibits outstanding robustness performance while maintaining acceptable image quality.

Table 9: Additional Image Quality Comparison.

|                | DwtDct [4] | SSL [20] | Stable Signature [19] | StegaStamp [46] | FreqMark |
|----------------|------------|----------|-----------------------|-----------------|----------|
| CLIP-FID [39]  | 2.36       | 6.88     | 1.70                  | 5.50            | 3.84     |
| L2             | 7.71       | 52.06    | 63.74                 | 85.24           | 52.95    |

**TPR/FPR Results on Diffusion Attacks**   We present the average TPR/FPR results across varying diffusion steps on the two datasets. Table 10 indicates that FreqMark maintains excellent performance in TPR@0.1% FPR, particularly at higher diffusion steps.

Table 10: The TPR results for different Diffusion Steps and FPR values.

| Diffusion Steps / FPR | 1.5e-2 | 1e-3 | 3e-5 | 3e-7 | 7e-10 | 1e-13 |
|:---:|:---:|:---:|:---:|:---:|:---:|:---:|
| 60 | 1.000 | 1.000 | 0.996 | 0.990 | 0.927 | 0.636 |
| 80 | 1.000 | 1.000 | 0.946 | 0.778 | 0.360 | 0.019 |
| 100 | 0.995 | 0.941 | 0.742 | 0.486 | 0.153 | 0.008 |
| 120 | 0.936 | 0.804 | 0.465 | 0.147 | 0.024 | 0.000 |
| 140 | 0.853 | 0.569 | 0.240 | 0.048 | 0.000 | 0.000 |
| 160 | 0.667 | 0.328 | 0.120 | 0.027 | 0.000 | 0.000 |
| 180 | 0.486 | 0.193 | 0.052 | 0.000 | 0.000 | 0.000 |
| 200 | 0.294 | 0.094 | 0.010 | 0.000 | 0.000 | 0.000 |

Table 11: Average Bit Accuracy on Gaussian noise and regeneration attacks for different combinations of Latent Noise and Pixel Noise.

| Latent Noise Std | Pixel Noise Std | | | | | | | | | | |
|:---:|:---:|:---:|:---:|:---:|:---:|:---:|:---:|:---:|:---:|:---:|:---:|
| | 0.00 | 0.02 | 0.04 | 0.06 | 0.08 | 0.10 | 0.12 | 0.14 | 0.16 | 0.18 | 0.20 |
| 0.00 | 0.706 | 0.749 | 0.797 | 0.803 | 0.751 | 0.758 | 0.715 | 0.706 | 0.678 | 0.694 | 0.701 |
| 0.05 | 0.718 | 0.759 | 0.800 | 0.804 | 0.804 | 0.767 | 0.781 | 0.723 | 0.730 | 0.744 | 0.704 |
| 0.10 | 0.768 | 0.793 | 0.870 | 0.865 | 0.858 | 0.820 | 0.816 | 0.817 | 0.794 | 0.801 | 0.780 |
| 0.15 | 0.841 | 0.850 | 0.897 | 0.908 | 0.891 | 0.907 | 0.889 | 0.890 | 0.877 | 0.861 | 0.832 |
| 0.20 | 0.850 | 0.878 | 0.934 | 0.910 | 0.910 | 0.878 | 0.867 | 0.868 | 0.869 | 0.870 | 0.840 |
| 0.25 | 0.889 | 0.915 | 0.924 | 0.926 | 0.896 | 0.885 | 0.885 | 0.852 | 0.866 | 0.860 | 0.820 |
| 0.30 | 0.820 | 0.853 | 0.886 | 0.912 | 0.894 | 0.885 | 0.857 | 0.847 | 0.840 | 0.863 | 0.830 |
| 0.35 | 0.791 | 0.838 | 0.867 | 0.866 | 0.865 | 0.872 | 0.805 | 0.811 | 0.785 | 0.804 | 0.814 |
| 0.40 | 0.762 | 0.802 | 0.865 | 0.868 | 0.814 | 0.835 | 0.802 | 0.807 | 0.768 | 0.763 | 0.754 |
| 0.45 | 0.725 | 0.777 | 0.846 | 0.822 | 0.824 | 0.767 | 0.764 | 0.782 | 0.767 | 0.742 | 0.738 |
| 0.50 | 0.719 | 0.763 | 0.814 | 0.838 | 0.809 | 0.787 | 0.763 | 0.731 | 0.727 | 0.733 | 0.707 |

**Noising Scale** Table 11 displays the average bit accuracy under Gaussian noise and three regeneration attacks for different combinations of Latent Noise and Pixel Noise. The results substantiate that judiciously introducing moderate noise in both the latent and pixel dimensions can effectively enhance the robustness of our method.

## A.4 Additional Qualitative Results

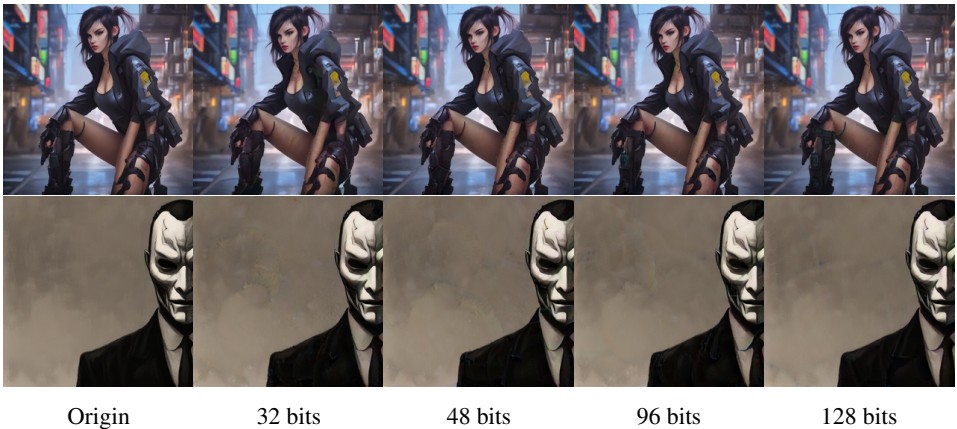

| Origin | 32 bits | 48 bits | 96 bits | 128 bits |

Figure 9: Examples of watermarked images with various encoding bits. FreqMark maintains image quality(approximately 31 dB in terms of PSNR) without degradation as the number of encoding bits increases while the bit accuracy against regeneration and Gaussian noise attacks may be reduced. Encoding 48 bits is the default setting.

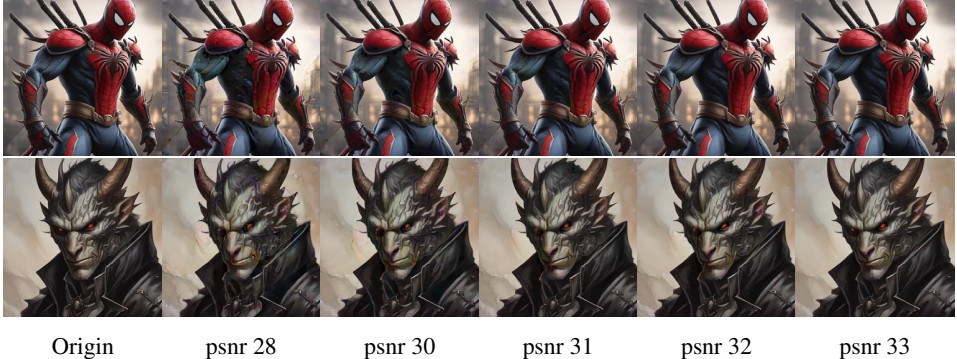

| Origin | psnr 28 | psnr 30 | psnr 31 | psnr 32 | psnr 33 |

Figure 10: Examples of watermarked images with various quality. FreqMark allows users to balance robustness and image quality based on their requirements. Due to the inherent robustness advantage of FreqMark, it still remains competitive at higher image quality settings.

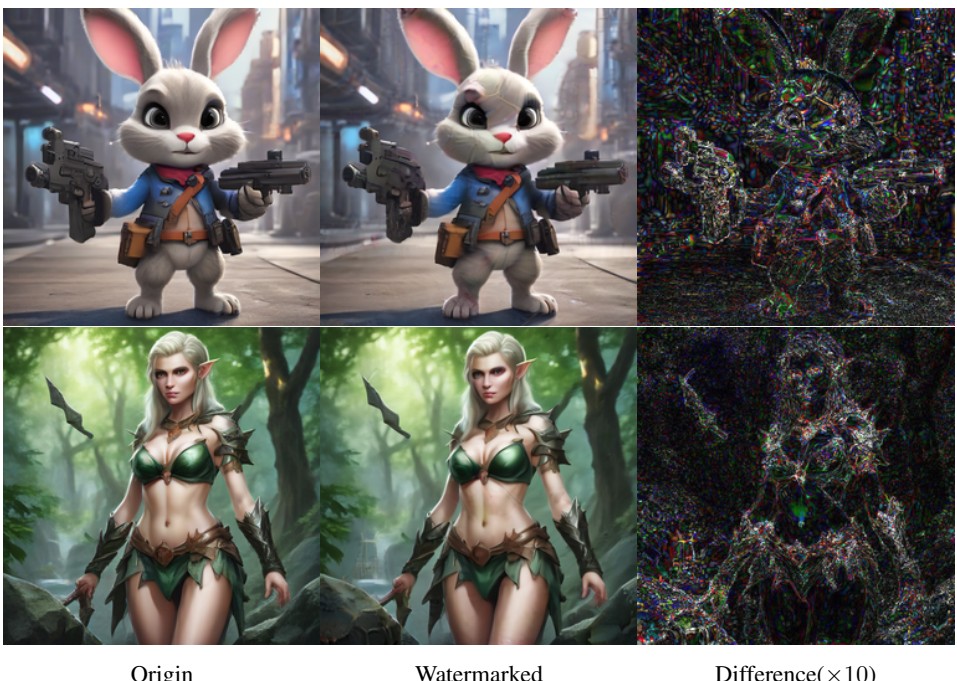

| Origin | Watermarked | Difference(×10) |

Figure 11: Two examples of poor image quality are presented: in the first image, a visible Moire-like pattern is introduced, while in the second image, subtle alterations to the semantic information in the background region are observed. This situation typically arises from the significant quality loss caused by the VAE during the image reconstruction process. Although this phenomenon is not frequent, future VAE models with improved performance are anticipated to mitigate this issue effectively. Moreover, this is a direction for further exploration and enhancement in future research.

| Original | Watermarked | Difference(×10) | Original | Watermarked | Difference(×10) |
|---|---|---|---|---|---|

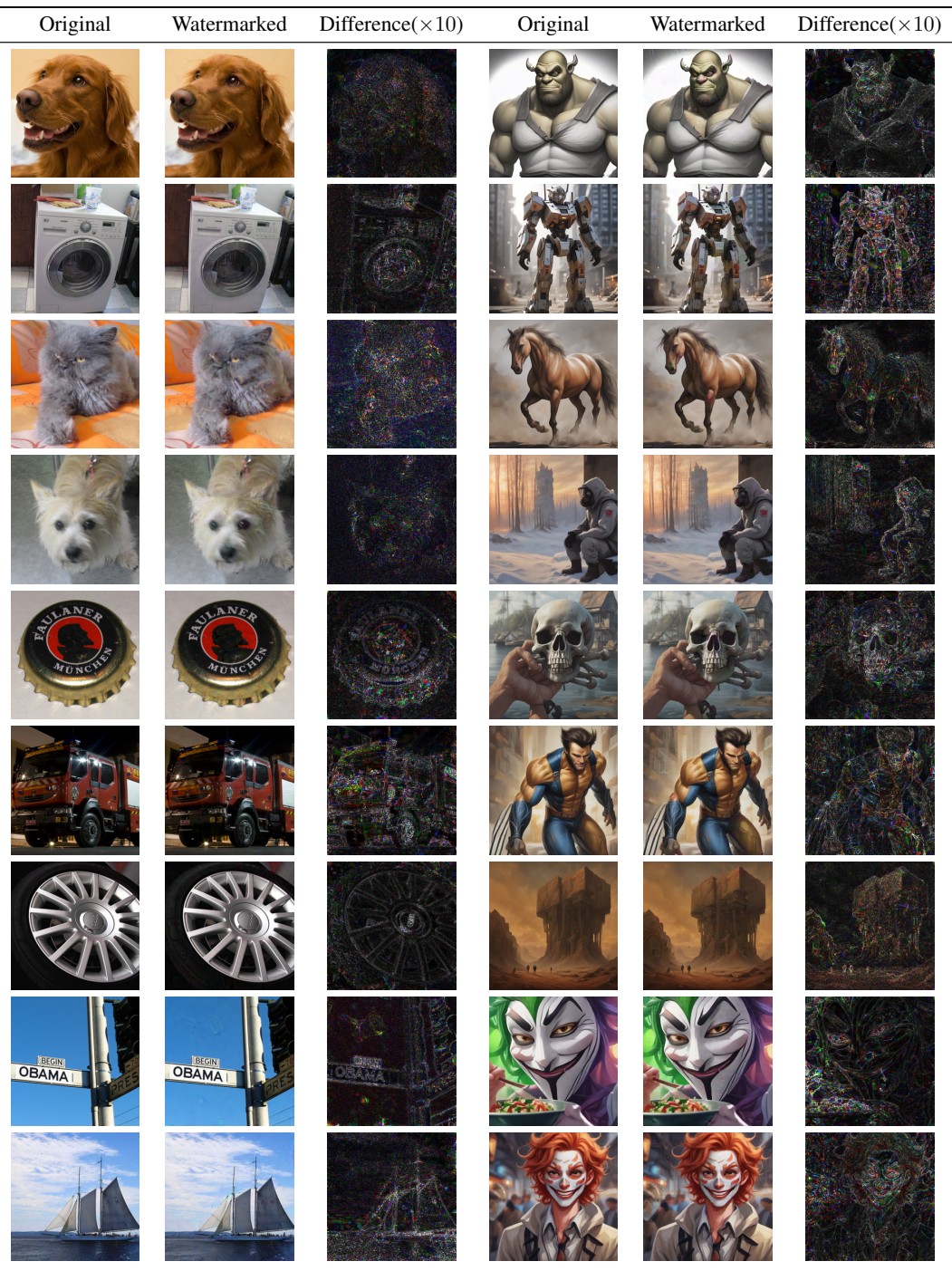

Figure 12: Additional qualitative results with a 48-bit hidden message. The images on the left are the results from ImageNet [16], while those on the right are from DiffusionDB [49].

