# OpenReview forum: "FreqMark: Invisible Image Watermarking via Frequency Based Optimization in Latent Space"
_NeurIPS.cc/2024/Conference — NeurIPS 2024 poster_

### Official Review · Reviewer_TJux · 2024-06-25

**Soundness:** 3
**Presentation:** 3
**Contribution:** 3
**Rating:** 6
**Confidence:** 4

**Summary:**

This paper considers the problem of watermarking of images. In particular, it introduces a method called FreqMark where watermark is embedded in the latent frequency space obtained after variation auto encoder (VAE) encoding. Numerical results are carried out on test datasets containing 500 randomly selected images from the ImageNet and 500 images from the DiffusionDB dataset.  Comparative results against DwtDctSvd, Stable Signature and SSL indicate that FreqMark exhibits superior quality-robustness tradeoff compared to the benchmark methods.

**Strengths:**

The main strengths of this paper are the novelty of the proposed FreqMark algorithm and the superior performance of FreqMark compared to the baseline image watermarking methods.

The main idea behind FreqMark is hiding the message bits in the FFT outputs of latent vectors coming from the VAE. In numerical experiments on two image datasets (ImageNet and Diffusion DB), authors show that FreqMark provides a bit accuracy of better than 90% for a 48-bit encoding setting. The Ablation Studies investigating image quality, number of encoding bits, noise levels and spatial perturbations are reasonably convincing of the benefits offered by FreqMark.

The Appendices provided contain significant additional information regarding FreqMark and is one of the strengths of this paper.

**Weaknesses:**

One of the main weaknesses of the work is that there is no convincing proof or theoretical justification of why hiding watermark bits in the FFT of latent space should be beneficial. The usual argument of using frequency domain for hiding watermark bits makes sense in that the FFT is applied to images (i.e., image pixels) and captures the spatial relationships between the image pixels. What kind of "spatial" relationship is captured by the FFT in the latent space and why is that beneficial for hiding watermark?

FFT produces complex values, yet there is no discussion of how the watermark bits are embedded on these complex values. In Fig. 1, the outputs of FFT are displayed as magnitude images and if this is what is intended, what happened to the phase of the FFT outputs.

**Questions:**

1. Are the FFT outputs complex and if so, how are the watermark bits being embedded?

2. What do FFT outputs represent when applied to the latent vectors? When applied to 1D time signals or 2D images, the FFT output describes the frequency content of the signals and images. What does the FFT of latent space represent and why is it useful?

**Limitations:**

Authors mention that there is room for optimization in the fusion of images and watermarks. It would have been useful to have a more detailed discussion of this aspect of the proposed method.

---

> ### Author Rebuttal · Authors · 2024-08-07
>
> Dear Reviewer TJux,
>
> We sincerely appreciate your thoughtful review and constructive suggestions on our paper. We have carefully considered your comments and have taken them into account.
>
> > W1: One of the main weaknesses of the work is that there is no convincing proof or theoretical justification of why hiding watermark bits in the FFT of latent space should be beneficial. The usual argument of using frequency domain for hiding watermark bits makes sense in that the FFT is applied to images (i.e., image pixels) and captures the spatial relationships between the image pixels. What kind of "spatial" relationship is captured by the FFT in the latent space and why is that beneficial for hiding watermark?
>
> **A1:** The primary distinction between the latent and pixel domains is that the latent domain is the result of information compression and downsampling of the pixel domain. Training on the pixel FFT alters the overall distribution of the image, making it more robust against common attacks. A similar effect occurs in the latent FFT domain. The key difference is that the compressed latent information is restored back to the pixel domain with the assistance of the model's prior knowledge through the VAE decoder. During this process, the watermark information added to the latent FFT interacts and fuses with the semantic information of the image. Consequently, FreqMark exhibits enhanced robustness against regeneration attacks, as detailed in Section 5.3. This also serves as the answer to question 2.
>
> *Bit Accuracy on DiffusionDB datasets:*
>
> | Bit Acc       | JPEG      | Gauss noise | VAE-B     | VAE-C     | Diffusion |
> | - | - | - | - | - | - |
> | Pixel Frequency      | 0.950     | **1.000**   | 0.797     | 0.775     | 0.596     |
> | Latent               | 0.906     | 0.804       | 0.796     | 0.833     | 0.675     |
> | **Latent Frequency** | **1.000** | 0.934       | **0.925** | **0.897** | **0.945** |
>
> > W2: FFT produces complex values, yet there is no discussion of how the watermark bits are embedded on these complex values. In Fig. 1, the outputs of FFT are displayed as magnitude images and if this is what is intended, what happened to the phase of the FFT outputs.
>
> **A2:** The output of latent FFT is complex, and we optimize both the real and imaginary parts during the optimization process.
>
>
> The latent FFT is then restored to the pixel domain through IFFT and the VAE decoder. The perturbation in the latent FFT domain alters the feature vector obtained after passing the image through an image encoder (DINO v2 in this case), thus achieving message encoding.
>
> For example, the output vector after perturbation for an image is [-2, 1, -0.5] . We predefine three vector groups as [1, 0, 0], [0, 1, 0], and [0, 0, 1] (assuming a 3-bit message encoding for simplicity). After performing element-wise multiplication, we obtain the results -2, 1, and -0.5. We use the sign of these results as the 0-1 bit encoding, ultimately yielding an encoded message of **010**. This also serves as the answer to question 1.
>
> > L1: Authors mention that there is room for optimization in the fusion of images and watermarks. It would have been useful to have a more detailed discussion of this aspect of the proposed method.
>
> **A3:** Experimental results demonstrate that the image quality post watermark addition via FreqMark does not significantly deteriorate compared to the image quality achieved solely through VAE encoding and decoding. Therefore, a VAE with enhanced performance may potentially further improve the quality of the watermarked images.
>
> | Method               | PSNR      | SSIM        |
> | -------------------- | --------- | ----------- |
> | FreqMark             | 31.20±1.54| 0.854±0.029 |
> | VAE Reconstruction   | 31.22±1.96| 0.879±0.032 |
>
> We plan to further optimize our method from the following aspects:
>
> 1. Exploring the necessity of optimizing all elements in the latent FFT could be a focus for further investigations. Achieving better integration might be possible by optimizing only a portion or a specific channel while maintaining performance.
>
> 2. The utilization of diffusion models' robust capabilities could potentially facilitate the implantation of watermarks, resulting in superior fusion effects.
>
> 3. With the assurance of performance, a strategy could be developed to further blend the watermarked image post FreqMark processing with the original image, potentially leading to enhanced fusion effects.
>
> We genuinely appreciate your valuable suggestions and will continue to optimize and explore our methods with enthusiasm.

---

> > ### Comment · Reviewer_TJux · 2024-08-12
> > **Response to Authors**
> >
> > Thank you for your response to my review. While your responses adequately answer my questions, I will keep my original rating as I see that other reviewers have raised other important concerns that need to be addressed.

---

> ### Author Response · Authors · 2024-08-13
>
> Thank you for your consideration! We have further addressed the concerns of other reviewers, which you can directly view.

---

### Official Review · Reviewer_kwvu · 2024-07-11

**Soundness:** 3
**Presentation:** 3
**Contribution:** 3
**Rating:** 6
**Confidence:** 4

**Summary:**

This paper introduces FreqMark, a novel invisible watermarking method that enhances digital content protection through optimization in the image's latent frequency space. Experiments have been conducted to demonstrate the robustness against regeneration attacks like VAE and diffusion model.

**Strengths:**

1. The paper is well-written and well-organized.

2. Extensive experiments have been conducted to demonstrate the robustness against various attacks including VAE and diffusion models.

**Weaknesses:**

1. Lacks some theoretical analysis. In the experiments, do you need to use the same VAE model for the attack as you have used during watermarking training? If not, then an analysis may be needed as to why it is still robust against attacks with different VAE models.

2. Since the proposed method requires case-by-case optimization, what is the watermarking time for each case, and how does it compare to other competing methods?

3. Why do we need a set of pre-trained N-dimensional direction vectors, but not directly produce the message?

4. In Equation (4), why $k \in N$? $N$ is a number, not a set, so $\in N$ is not appropriate. Besides, using $v^N$ to denote a vector set and $v^k$ to represent a vector can lead to confusion. In Figure 2, there are $K+1$ vectors in the set, I think $K+1$ is the bit length.

4. Minors:
- $\epsilon_1$ and $\epsilon_2$ (or $\varepsilon_1$ and $\varepsilon_2$) appear in Figure 2, so it is better to mention them in the caption of Figure 2.
- In line 213, page 7, the word 'We' should be 'we'.

**Questions:**

Please see the Weaknesses.

**Limitations:**

The authors have addressed the limitations in their paper.

---

> ### Author Rebuttal · Authors · 2024-08-07
>
> Dear Reviewer kwvu,
>
> Thank you very much for your careful reading and recognition of our work, we will address each one sequentially below:
>
> > W1: Lacks some theoretical analysis. In the experiments, do you need to use the same VAE model for the attack as you have used during watermarking training? If not, then an analysis may be needed as to why it is still robust against attacks with different VAE models.
>
> **A1:** In the experiment, it is not necessary to use the same VAE model for attacks as the one used for watermark training.
>
> Essentially, adding information will inevitably alter some aspects of the image, and our goal is to ensure that this information remains as intact as possible during attacks. As demonstrated in Section 5.3 and Table 3, different optimization methods exhibit distinct characteristics.
> - Directly embedding watermarks in the Pixel or Latent domains makes them highly susceptible to various traditional attacks, as specific bits of information may independently reside in one or more pixels. In contrast, embedding watermarks in the frequency domain effectively resists multiple traditional attacks because this method distributes the information across the entire image.
> - Similarly, watermarks embedded in the pixel space are weak against regeneration attacks, whereas embedding watermarks in the latent space significantly enhances robustness against such attacks. This is due to the stronger correlation between the optimized latent embeddings and the image semantics, making the watermark more resistant to disruption by regeneration attacks.
>
> FreqMark embeds watermarks in the latent frequency space, combining these two aspects to produce a synergistic effect, making FreqMark robust against traditional attacks and regeneration attacks.
>
> Furthermore, due to FreqMark's characteristic of training the image itself, it exhibits stronger robustness against regeneration attacks using the same VAE.
>
> *Bit Accuracy of regeneration attack using the same VAE*
>
> | PSNR after VAE attack | 31.43 | 30.31 | 28.98 | 27.39 | 25.82 |
> | - | - | - | - | - | - |
> | Bit Acc | 1.000 | 1.000 | 0.998 | 0.990 | 0.975 |
>
> > W2: Since the proposed method requires case-by-case optimization, what is the watermarking time for each case, and how does it compare to other competing methods?
>
> **A2:** We employed half-precision training to reduce watermarking time and GPU memory usage without compromising performance. In our current experiments, using a single A-100 GPU with 40GB memory, and process four 512x512 images in parallel for 400 steps takes about 0.75 minutes per image. Increasing the batch size will improve overall efficiency.
>
> As shown in Figure 7 in the Appendix, FreqMark demonstrates considerable performance at just 200 steps, and even at 100 steps. This allows for the reduction of steps as needed to save time. According to our experiments, processing four 512x512 images in parallel for 100 steps can be accelerated to about 12 seconds per image.
>
> *Bit Accuracy with different number of steps*
> | Bit Acc | JPEG | Gauss noise | VAE-B | VAE-C | Diffusion |
> | - | - | - | - | - | - |
> | 400 steps |1.000 | 0.934 |0.925 |0.897 |0.945 |
> | 200 steps |1.000 | 0.930 |0.923 |0.885 |0.933 |
> | 100 steps |0.987 | 0.922 |0.921 |0.881 |0.888 |
>
> To train the Stable Signature, one must first spend a day training the watermark extractor on 8 GPUs. Subsequently, for any specific hidden message, the stable signature requires approximately 1 minute to fine-tune the VAE decoder.
>
> SSL directly optimizes the image pixels, resulting in faster processing speed. Under the same experimental settings, it takes approximately 1 second to process per image. However, FreqMark demonstrates a significantly stronger robustness advantage than SSL.
>
> We will further attempt to reduce the optimization time without compromising performance.
>
>
> > W3: Why do we need a set of pre-trained N-dimensional direction vectors, but not directly produce the message?
>
> **A3:** The method of extracting hidden watermark information by predefined vector directions is referenced from SSL. Overall, this approach is more suitable for post-generation self-supervised methods, providing stronger robustness for hidden watermarks while maintaining image quality. Additionally, this method effectively addresses the challenge of obtaining hidden watermark information without the need for additional training of the extraction network.
>
> > W4: In Equation (4), why $k∈N$? N is a number, not a set, so $∈N$ is not appropriate. Besides, using $v^N$ to denote a vector set and $v^k$ to represent a vector can lead to confusion. In Figure 2, there are $K+1$ vectors in the set, I think $K+1$ is the bit length.
>
> Thank you for your careful reading and for pointing out the errors. This section of the paper indeed has some incorrect expressions that caused confusion, and we will make the necessary corrections.
>
> The dimension of the vector is N-dimensional. For clarity and expression, the number of pre-defined vectors is defined as K (we will correct Vector 0 to Vector 1 in Figure 2). To avoid interference between different bits of information, the pre-defined vectors should be orthogonal to each other, thus K should be less than or equal to N. This means that a maximum of N bits of information can be embedded for an N-dimensional vector.
>
> Therefore, The correct expression should be that the pre-defined vectors are a set of K N-dimensional vectors $V^N = \\{v^1, v^2, ..., v^K \mid K\leq N\\}$.
>
>  And Equation 4 should be corrected to:
>
> $m_d^k = sign(z_{I_{w}} \cdot v^k) = sign(E_{img}(I_{w}) \cdot v^k), v^k \in V^N$
>
> We will continue to carefully review and correct any mistaken and confused statements in the paper.
>
> Thanks again for your suggestions and corrections!

---

> > ### Comment · Reviewer_kwvu · 2024-08-12
> >
> > Thanks for your response. My concerns are addressed. I will keep my rating.

---

> > > ### Author Response · Authors · 2024-08-13
> > >
> > > Thank you for your feedback and consideration!

---

### Official Review · Reviewer_BgZJ · 2024-07-12

**Soundness:** 3
**Presentation:** 2
**Contribution:** 2
**Rating:** 4
**Confidence:** 3

**Summary:**

This paper proposes a method called FreqMark that is able to prevent the invisible watermarks from the regeneration attack. By using the unconstrained optimization of the image latent frequency space obtained after VAE encoding, the proposed FreqMark achieves better robustness against the regeneration attacks and traditional attacks.

**Strengths:**

The proposed method achieves better robustness against regeneration attacks. By using a newly proposed optimization strategy, the proposed method achieves a balance between the image quality and the envisioned robustness.

**Weaknesses:**

1. The major contribution of this paper is to introduce a kind of image watermark robust to regeneration attacks. However, this part is not highlighted in its title. The authors are suggested to consider this in the title.
2.  The proposed methodology follows an established manner with incremental innovations. The whole approach does not demonstrate enough differences when compared with the previous approaches.
3. For the second contribution, the authors claim that their proposed framework is flexible and then say such flexibility guarantees a trade-off between the bits number, image quality, and robustness. However, why can such properties be considered as flexibility?
4. The whole pipeline presented in Figure 2 lacks novelty. This is just a very common encoder and decoder framework. The authors need to better consider their contributions for this part.

**Questions:**

I have shown my concerns in the weakness part. The authors are suggested to consider them during the rebuttal.

**Limitations:**

Please address my concerns within the weakness part.

---

> ### Author Rebuttal · Authors · 2024-08-07
>
> Dear Reviewer BgZJ,
>
> Thank you for your insightful feedback and questions. We address the concerns as follows:
>
> > W1: The major contribution of this paper is to introduce a kind of image watermark robust to regeneration attacks. However, this part is not highlighted in its title. The authors are suggested to consider this in the title.
>
> **A1:** Thank you for your suggestion, we will consider it positively. Yes, FreqMark's ability to resist regeneration attacks is a significant advantage, and we believe the primary contribution of the paper lies in the innovative approach of embedding watermark in the latent frequency space, and the extensive experiments conducted to demonstrate its effectiveness.
>
> > W2: The proposed methodology follows an established manner with incremental innovations. The whole approach does not demonstrate enough differences when compared with the previous approaches.
>
> **A2:** The core idea of FreqMark lies in embedding watermarks in the latent frequency space of images. Frequency domain optimization subtly alters the overall image, enabling the watermark information to be hidden within the entire picture. The latent domain represents the compressed image, and during the process of restoring the image using the decoder, the added watermark information naturally integrates with the semantic information of the image. This results in outstanding robustness when facing regeneration attacks and various other types of attacks.
>
> *Bit Accuracy on both ImageNet and DiffusionDB datasets:*
>
> | Bit Acc | JPEG | Gauss noise | VAE-B | VAE-C | Diffusion |
> | - | - | - | - | - | - |
> | SSL | 0.964 | **0.946** | 0.768 | 0.761 | 0.736 |
> | Stable Signature | 0.864 | 0.930 | 0.6490 | 0.689 | 0.531 |
> | **FreqMark** | **0.996** | 0.937 | **0.932** | **0.911** | **0.957** |
>
> Furthermore, our experiments demonstrate the impact of different watermark embedding positions on robustness, and we delve into the advantages and limitations of FreqMark.
>
> *Bit Accuracy on DiffusionDB datasets:*
>
> | Bit Acc | JPEG | Gauss noise | VAE-B | VAE-C | Diffusion |
> | - | - | - | - | - | - |
> | Pixel Frequency | 0.950 | **1.000** | 0.797 | 0.775 | 0.596 |
> | Latent | 0.906 | 0.804 | 0.796 | 0.833 | 0.675 |
> | **Latent Frequency** | **1.000** | 0.934 | **0.925** | **0.897** | **0.945** |
>
> FreqMark also boasts several additional advantages. It is a plug-and-play solution that can be directly applied to existing pre-trained models without any network training. During the watermark embedding process, users can freely adjust parameters to suit their specific needs, offering considerable flexibility, which will be discussed in detail in the W3.
>
> It is important to emphasize that FreqMark differs from previous methods.
> - FreqMark, as a post-generation method, is fundamentally distinct from merged-in-generation methods like Stable Signature, and it offers greater convenience and flexibility in use.
> - Unlike most works such as StegaStamp, FreqMark does not require the training of any network parameters.
> - Compared to other network-training free methods like SSL, FreqMark substantially enhances robustness through its unique watermark embedding approach.
>
> > W3: For the second contribution, the authors claim that their proposed framework is flexible and then say such flexibility guarantees a trade-off between the bits number, image quality, and robustness. However, why can such properties be considered as flexibility?
>
> **A3:** There are three crucial attributes for invisible image watermarking—bits number, image quality, and robustness. Based on information theory, these attributes form an impossible trinity, meaning it is impossible to achieve all three simultaneously. For instance, if a method achieves high bits number and image quality, it inevitably cannot achieve high robustness; similarly, if it aims for high bits number and robustness, it cannot maintain high image quality. Thus, users need to make trade-offs among these attributes based on different application scenarios.
>
> Specifically, the flexibility of FreqMark is reflected in the following tow aspects:
>
> 1. Merged-in-generation methods fix the model's capabilities during the training process, which prevent users from choosing between three attributes. FreqMark grants this choice to users, allowing them to freely adjust parameters during the watermark embedding process to achieve the desired balance.
> 2. Users can independently choose the VAE model and image encoder. Additionally, the pre-defined vector can be customized by users, serving as a key that adds an extra layer of security to the watermark.
>
> > W4: The whole pipeline presented in Figure 2 lacks novelty. This is just a very common encoder and decoder framework. The authors need to better consider their contributions for this part.
>
> **A4:** Figure 2 illustrates the entire process of watermark embedding and extraction, while the main idea of the paper is depicted in Figure 1. We still believe that presenting the overall process is necessary, as it helps readers better understand our method. Although Figure 2 reflects the encoder-decoder framework, it differs from other methods in the following ways:
>
> 1. FreqMark uses only pre-trained encoders and decoders without requiring any network training. In Figure 2, we use a "snowflake" icon to denote the frozen neural networks.
> 2. FreqMark achieves watermark embedding by training perturbations in the latent frequency space. In Figure 2, we use a "fire" icon to indicate the trained perturbations.
>
> Thank you for your suggestion. We will consider improving or adding image to better highlight our contributions.

---

> > ### Comment · Reviewer_BgZJ · 2024-08-13
> >
> > After reading the rebuttal, some of my concerns about the novelty are still not effectively addressed. Though I know that it may not be a very good way to use this for the judgment of a work, this is still my concern this time. I keep my original scores while I respect the other reviewers' and AC's decisions. Thanks.

---

> > > ### Author Response · Authors · 2024-08-13
> > >
> > > Dear Reviewer BgZJ,
> > > Thank you again for your suggestions and time. We would like to further elaborate on the novelty of our approach.
> > >
> > > > The proposed methodology follows an established manner with incremental innovations. The whole approach does not demonstrate enough differences when compared with the previous approaches.
> > >
> > > FreqMark is distinct from previous methods and has the following advantages:
> > >
> > > - Unlike classical encoder-decoder network framework such as StegaStamp, FreqMark does not require training any network parameters, demonstrating a notable distinction.
> > > - FreqMark showcases the value of latent FFT space, exhibiting substantial performance advantages compared to pixel space.
> > > - FreqMark demonstrates strong robustness while maintaining image quality and provides users with sufficient flexibility to accommodate diverse needs.
> > >
> > > Furthermore, additional experimental results also serve as evidence for FreqMark's superior performance.
> > >
> > > 1. FreqMark exhibits high robustness even under intense VAE regeneration attacks.
> > >
> > > |PSNR after Vae Attack|31.43|30.31|28.98|27.39|25.82|
> > > |-|-|-|-|-|-|
> > > |Bit Acc|1.000|1.000|0.998|0.990|0.975|
> > >
> > > 2. When facing stronger diffusion attacks, FreqMark maintains a high level of TPR@0.1%FPR.
> > >
> > > |Diffusion Steps/FPR|1.5e-2|1e-3|3e-5|3e-7|7e-10|1e-13|
> > > |-|-|-|-|-|-|-|
> > > |60|1.000|1.000|0.996|0.990|0.927|0.636|
> > > |80|1.000|1.000|0.946|0.778|0.360|0.019|
> > > |100|0.995|0.941|0.742|0.486|0.153|0.008|
> > > |120|0.936|0.804|0.465|0.147|0.024|0.000|
> > > |140|0.853|0.569|0.240|0.048|0.000|0.000|
> > > |160|0.667|0.328|0.120|0.027|0.000|0.000|
> > > |180|0.486|0.193|0.052|0.000|0.000|0.000|
> > > |200|0.294|0.094|0.010|0.000|0.000|0.000|
> > >
> > > 3. FreqMark also demonstrates great performance when facing the adversarial attacks target latent representations.
> > >
> > > $$max_{x_{adv}}|f(x_{adv}) - f(x)\|_2$$
> > >
> > > $$s.t. \|x_{adv} - x\|_\infty \leq \epsilon$$
> > >
> > > |Attack Strength(eps)|Bit Acc|TPR@0.1%FPR|
> > > |-|-|-|
> > > |2/255|1.000|1.000|
> > > |4/255|0.987|1.000|
> > > |6/255|0.944|0.986|
> > > |8/255|0.893|0.972|
> > >
> > > We hope the additional experimental results could address your concerns, and we appreciate your valuable feedback!

---

### Official Review · Reviewer_BZ7h · 2024-07-15

**Soundness:** 3
**Presentation:** 3
**Contribution:** 1
**Rating:** 6
**Confidence:** 4

**Summary:**

The authors propose a new post-processing watermark for imagery, FreqMark. The FreqMark embeds a binary message into the frequency domain of a VAE-encoded image via a small perturbation, and then following IFFT + decompression, utilizes a pre-trained image encoder to extract the message. A PSNR + LPIPS metric is used to protect mage quality against the message embedding. The dual approach of VAE encoding + Gaussian noise makes the mark resilient against regeneration and Gaussian noise attacks. The method demonstrates high bit accuracy and good image quality when compared against DwtDct, SSL, Stable Signature on watermarked DiffusionDB & Imagenet images.

**Strengths:**

-Easy-to-follow narrative and nice motivation in the age of generative imagery.

The intuition behind the FreqMark algorithm is sound, easy to follow, and presumably simple to implement.

-Method demonstrates robustness against regeneration, which is a very potent attack.

**Weaknesses:**

This work is missing two critical competitor baselines which must be addressed.

-In my opinion, the FreqMark is an incremental variant of the StegaStamp [1]. Surprisingly, the authors did not compare against this method even though it is well-known in existing literature that the StegaStamp is robust against many attacks, including regeneration [2]. Like FreqMark, the StegaStamp increases resilience against attacks by incorporating them into the training pipeline and uses a critic loss to preserve image quality (LPIPS + L2, versus LPIPS + PSNR for FreqMark). The resemblance of equation (10) in this manuscript to the loss function equation (2) in [1] begs the question of novelty. It is also incorrectly stated in line 71 that the StegaStamp only relies on differential image perturbations for the training pipelines -- in fact, any attack can be added, as the decoder is trained after image manipulation.

-Again, if the spirit of the paper is to increase resilience against regenerations, the authors also needed to compare against the state-of-the-art Tree-Ring watermark [4], which was noted to be incredibly resilient against regenerations by the authors of the regeneration attack [2]. As the Tree-Ring is an in-processing technique that embeds a message within a diffusion process, one way to set up a comparison is to post-process a collection of Tree-Ring watermarked images via FreqMark, and then independently extract both watermarks.

-As noted in [3], there is no single perceptual metric that is an objective measure of image quality, thus low PSNR or LPIPS distance does not necessarily indicate the method is not introducing artifacts. The authors need to add 1-2 more metrics (maybe L2 and FID, for example) for a more convincing argument.

-500 images is too small a sample size for the tested FPR thresholds. Modern literature in this field such as [3,4,5] are using several thousand images.

-VAE regenerations are far weaker compared to diffusion regenerations. Readers will want to see how the FreqMark holds up against longer, deeper regenerations (>= 100 steps) to see how the decoding accuracy is affected.

-As observed in [3], the use of publicly available VAEs to encode/decode watermarks is easily defeated if the attacker uses a regeneration leveraging encoders/decoders with the same architecture.


[1] Tancik, M., Mildenhall, B., & Ng, R. (2020). Stegastamp: Invisible hyperlinks in physical photographs. In Proceedings of the IEEE/CVF conference on computer vision and pattern recognition (pp. 2117-2126).

[2] Zhao, X., Zhang, K., Su, Z., Vasan, S., Grishchenko, I., Kruegel, C., ... & Li, L. (2023). Invisible image watermarks are provably removable using generative ai. arXiv preprint arXiv:2306.01953.

[3] An, B., Ding, M., Rabbani, T., Agrawal, A., Xu, Y., Deng, C., ... & Huang, F. (2024). Benchmarking the robustness of image watermarks. arXiv preprint arXiv:2401.08573.

[4] Wen, Y., Kirchenbauer, J., Geiping, J., & Goldstein, T. (2023). Tree-ring watermarks: Fingerprints for diffusion images that are invisible and robust. arXiv preprint arXiv:2305.20030.

[5] Saberi, M., Sadasivan, V. S., Rezaei, K., Kumar, A., Chegini, A., Wang, W., & Feizi, S. (2023). Robustness of ai-image detectors: Fundamental limits and practical attacks. arXiv preprint arXiv:2310.00076.

**Questions:**

1. See weaknesses.

2. Which version of Stable Diffusion was used for the regeneration attack?

**Limitations:**

Adequately discussed.

---

> ### Author Rebuttal · Authors · 2024-08-07
>
> Dear Reviewer BZ7h,
> Thank you for your suggestions. We address the concerns as follows:
>
> > W1: In my opinion, the FreqMark is an incremental variant of the StegaStamp [1]. ... The resemblance of equation (10) in this manuscript to the loss function equation (2) in [1] begs the question of novelty.
>
> **A1:** In fact, there are significant differences between FreqMark and StegaStamp, and FreqMark is not an incremental variant of StegaStamp. We have discussed StegaStamp in the Related Work section and elaborated on the differences between the two kind of approaches in **lines 22 to 27 of the Introduction section**.
> - FreqMark leverages the representational capacity of pre-trained networks to optimize **the latent FFT of images**, without the need for training any additional networks, adopting a case-by-case watermarking  approach. In contrast, StegaStamp involves training an encoder and decoder under various perturbation augmentations.
> - FreqMark only incorporates two types of noise as augmentations during the training of image perturbations, it maintains excellent robustness under various attacks.
> - The loss functions of FreqMark and StegaStamp hold distinct meanings. FreqMark employs an image encoder to encode the image into a representative vector, which is then multiplied by a predefined vector group and the watermark image's representative vector. The encoding is determined by the positive or negative nature of the result, which is entirely different from StegaStamp.
>
> We conducted a comparison under the same bit number(100) and essentially consistent image quality.
>
> |Bit Acc|PSNR|SSIM|None|Brightness|Contrast|Jpeg|Blur|Noise|Vae-b|Vae-c|Diffusion|Avg|
> |-|-|-|-|-|-|-|-|-|-|-|-|-|
> |StegaStamp|28.47 |0.91| 0.999 | 0.999 | 0.998 | 0.994 | 0.997 | 0.991 | 0.981 | 0.984 | 0.857 | 0.978 |
> |FreqMark  |28.83 |0.88|1.000  |1.000|0.999|0.999|0.998|0.989| 0.976|0.934 |0.933|0.981|
>
> The findings indicate that FreqMark presents a balanced performance and exceptional robustness against diffusion attacks. Additionally, its network independence provides great flexibility and security. Users can dynamically choose the optimal image quality and encoding bit number, and customize their decoding vector group, thereby augmenting the security of the watermark decoding process.
>
> > W2: The authors also needed to compare against the state-of-the-art Tree-Ring watermark [4].
>
> **A2:** Tree-Ring integrates the watermarking process and generation process, encoding 1-bit message (with or without watermark) through its approach, which is an in-process method. FreqMark is a post-processing watermarking technique capable of encoding multi-bit message, thus offering a broader range of applicability and a higher amount of encoded message. Therefore, following the metric setup of Tree-Ring, we compared the TPR@1%FPR of FreqMark and Treering under various attacks.
> |TPR@1%FPR|None|Brightness|Contrast|Jpeg|Blur|Noise|Vae-b|Vae-c|Diffusion|Avg|
> |-|-|-|-|-|-|-|-|-|-|-|
> |Tree-Ring|1.000|1.000|1.000|0.996|0.999|0.918|0.991|0.995|0.998|0.989|
> |FreqMark|1.000|1.000|1.000|1.000|1.000|0.986|0.989|0.975|1.000|0.994|
>
>
> The experimental results demonstrate that the robustness performance of both methods is very close. However, it is important to note that FreqMark encode 48-bit message, which is a significantly larger amount of information compared to Treering's 1-bit encoding.
>
> > W3: As noted in [3], there is no single perceptual metric that is an objective measure of image quality...
>
> **A3:** Thank you for your suggestion. We have added CLIP-FID and L2 as new image quality metrics, with the results as follows:
>
> ||DwtDct|SSL|Stable Signature|StegaStamp|FreqMark|
> |-|-|-|-|-|-|
> |CLIP-FID|2.36|6.88|1.70|5.50|3.84|
> |L2|7.71|52.06|63.74|85.24|52.95|
>
> The experimental conclusions regarding image quality are consistent with the results in the paper, demonstrating that FreqMark exhibits outstanding robustness performance while maintaining acceptable image quality.
>
> > W4: 500 images is too small a sample size for the tested FPR thresholds.
>
> **A4:** We utilized 5,000 watermarked images to plot the FPR-TPR curve, and some key results are as follows:
>
> ||1e-3|3e-5|3e-7|7e-10|1e-13|
> |-|-|-|-|-|-|
> |Jpeg|1.000|0.999|0.992|0.989|0.978|
> |Noise|0.983|0.979|0.934|0.851|0.575|
> |Vae-b|0.987|0.967|0.935|0.819|0.484|
> |Vae-c|0.971|0.961|0.891|0.689|0.385|
> |Diffusion|1.000|0.996|0.990|0.927|0.636|
>
> The experimental results are close to those in the main text. We will update the experimental results in the paper accordingly.
>
> > W5: VAE regenerations are far weaker compared to diffusion regenerations.
>
> **A5:**  We have gradually increased the diffusion steps. The results are as follows:
>
> |Steps|60|80|100|120|140|160|180|200|
> |-|-|-|-|-|-|-|-|-|
> |Bit Acc|0.935|0.863|0.831|0.754|0.712|0.692|0.660|0.637|
>
> It can be seen that FreqMark maintains good robustness even at higher diffusion steps.
>
> > W6: As observed in [3], the use of publicly available VAEs to encode/decode watermarks is easily defeated...
>
> **A6:** The characteristic of FreqMark that trains the image itself rather than the network can ensure strong robustness when facing regeneration attacks using the same VAE.
>
> |PSNR after Vae Attack|31.43|30.31|28.98|27.39|25.82|
> |-|-|-|-|-|-|
> |Bit Acc|1.000|1.000|0.998|0.990|0.975|
>
> This is because the perturbation on the latent FFT changes the overall distribution of the image latent, making the watermark message affect the entire image globally. Therefore, it is difficult for perturbations on the image's latent to damage the watermark message. A similar phenomenon is observed in the pixel FFT of the image.
>
> |PSNR after Gaussian Noise Disrupt in Pixel Domain|31.09|29.68|28.04|26.46|25.04|
> |-|-|-|-|-|-|
> |Bit Acc|1.000|1.000|1.000|1.000|1.000|
>
> > Q1: Which version of Stable Diffusion was used for the regeneration attack?
>
> **A7:** Following the experimental setup of [2], we employ stable diffusion 2-1 to conduct diffusion attacks.

---

> ### Comment · Reviewer_BZ7h · 2024-08-12
> **Reply to authors**
>
> First I want to thank the other reviewers and authors for their detailed replies. Two items:
> 1. Another reviewer also questioned the motivation of using the FFT in the latent space. I am still not completely convinced by the author response as there are many adversarial attacks which specifically target the latent representation (see Adv-Emb attacks in [3].
> 2. Could the authors provide the TPR/FPR for the varied diffusion steps?
>
> The new comparisons against Tree-Ring (another FFT watermark) and StegaStamp highlight the utility of FreqMark. I am increasing my score to a 5, and looking forward to further response.

---

> > ### Author Response · Authors · 2024-08-13
> >
> > Dear Reviewer BZ7h,
> > Thank you again for your valuable feedback and time. We address the concerns as follows:
> >
> > > Q1: Another reviewer also questioned the motivation of using the FFT in the latent space. I am still not completely convinced by the author response as there are many adversarial attacks which specifically target the latent representation (see Adv-Emb attacks in [3].
> >
> > **A1:** Following the settings in [3], we applied adversarial attacks targeting the latent representations of the watermarked images, and the results are as follows:
> >
> > $$max_{x_{adv}}|f(x_{adv}) - f(x)\|_2$$
> >
> > $$s.t. \|x_{adv} - x\|_\infty \leq \epsilon$$
> >
> > |Attack Strength(eps)|Bit Acc|TPR@0.1%FPR|
> > |-|-|-|
> > |2/255|1.000|1.000|
> > |4/255|0.987|1.000|
> > |6/255|0.944|0.986|
> > |8/255|0.893|0.972|
> >
> > Experimental results demonstrate that FreqMark exhibits strong robustness when facing adversarial attacks targetinmg latent representations. We believe that this can be attributed to the limited impact of attacks targeting latent representations on the latent FFT domain.
> >
> > > Q2: Could the authors provide the TPR/FPR for the varied diffusion steps?
> >
> > **A2:** The TPR/FPR results for different diffusion steps are as follows:
> >
> > |Diffusion Steps/FPR|1.5e-2|1e-3|3e-5|3e-7|7e-10|1e-13|
> > |-|-|-|-|-|-|-|
> > |60|1.000|1.000|0.996|0.990|0.927|0.636|
> > |80|1.000|1.000|0.946|0.778|0.360|0.019|
> > |100|0.995|0.941|0.742|0.486|0.153|0.008|
> > |120|0.936|0.804|0.465|0.147|0.024|0.000|
> > |140|0.853|0.569|0.240|0.048|0.000|0.000|
> > |160|0.667|0.328|0.120|0.027|0.000|0.000|
> > |180|0.486|0.193|0.052|0.000|0.000|0.000|
> > |200|0.294|0.094|0.010|0.000|0.000|0.000|
> >
> > It can be observed that FreqMark also demonstrates great performance in terms of TPR@0.1%FPR at higher diffusion steps.
> >
> > Thank you again for your valuable suggestions. We will incorporate these results in the next version.

---

> > > ### Comment · Reviewer_BZ7h · 2024-08-13
> > > **Reply to authors**
> > >
> > > Thanks for these follow-up experiments demonstrating robustness against embedding attacks and the TPR of the FreqMark for varying diffusion lengthens. At 100 steps the method appears to be fairly robust (this is the point at which users may see visible artifacts). Please do investigate why there’s such a drastic reduction in TPR and bit acc from 100->120 steps. The progression of artifacts from 100->120 should not be significant to warrant this near exponential watermark degradation. You could look over the image quality metrics and see if these are dropping off steeply as well.
> > >
> > > In any case, I am increasing my score to a 6 and will deliberate with the AC and other reviewers.

---

> > > > ### Author Response · Authors · 2024-08-14
> > > >
> > > > Dear Reviewer BZ7h,
> > > >
> > > > Thank you for your patience and valuable suggestions, which have greatly contributed to the improvement of our work.
> > > >
> > > > We supplement the PSNR between the watermarked images after different diffusion step attacks and the original watermarked images.
> > > >
> > > > |Steps|60|80|100|120|140|160|180|200|
> > > > |-|-|-|-|-|-|-|-|-|
> > > > |Bit Acc|0.935|0.863|0.831|0.754|0.712|0.692|0.660|0.637|
> > > > |PSNR |27.67|26.95|26.19|25.46|24.92|24.47|23.99|23.57|
> > > >
> > > > The reduction could be due to the continuous decline in image quality reaching a certain critical threshold, causing the original watermark message to be damaged and leading to a significant decrease in bit accuracy (a drop of 0.077 at 100->120, and a drop of 0.042 at 120->140). Simultaneously, the decrease in PSNR at 120 steps is larger than at 140 steps.
> > > >
> > > > The TPR is more sensitive to changes in bit accuracy. Under the same FPR setting, a lower bit accuracy might cause a significant decrease in TPR.
> > > >
> > > > Thank you for your feedback and consideration again！

---

### Author Response · Authors · 2024-08-12

Dear reviewers:

Thanks again for your insightful suggestions and comments! According to your helpful comments, we have further supplemented our study with additional experimental results and more detailed descriptions to demonstrate the advantages and robustness of FreqMark as a network-free approach. More details and discussions are shown in the rebuttal parts. We will add the discussion below to our next version and be more than happy to answer any further questions.

---

### Decision · Program_Chairs · 2024-09-25

**Decision:**

Accept (poster)

**Comment:**

The paper has received four reviews, with three recommendations weakly recommending to accept the paper (6,6,4->6) and one weakly recommending to reject it (4)

The more critical review recognizes the improved robustness of the approach to regeneration attacks.  They are critical of the level of novelty in the technical contribution, considering it to be yet another variant of an encoder-decoder architecture.  They also query the motivation for flexibility in trading off the capacity, quality and robustness.

As the authors note, it is a well-known tradeoff in watermarking approaches to trade between capacity, perceptibility and robustness and enabling control over this is useful given different requirements of different use cases.  Therefore I would consider this a reasonable feature to highlight in the work.  Regarding the novelty, the authors have responded on this point.  I think it is reasonable that most modern ML watermarking approaches are some form of encoder-decoder approach – it appears the contribution here is actually to work with the latent spatial representation (running an optimization to perturb the 2D FFT of that) to introduce a mark.  It seems novel to my knowledge although I would agree not a huge contribution, it is an interesting take on the problem provided the analysis is done well enough to show a utility in the study.

The more positive reviews weakly recommend acceptance but have a number of criticisms.

BZ7h is concerned that the method is similar to StegaStamp which is another encoder-decoder approach that many papers will baseline against.  I can see that view but I think it is the FFT perturbation aspect of this work in the latent that distinguishes it.  The more serious criticism of this review is that StegaStamp and TreeRing are not compared against (particularly given TreeRing is a frequency method) and that primarily VAE regeneration is being considered as an attack, given the growing use of diffusion / img2img approaches.  That said, the authors provide the comparison in the rebuttal which helps mitigate these important shortfallings. The reviewer has moved their score to 5 and then to 6 ultimately in later discussions.  I feel this reviews critcisms were well addressed in the end.

kwvu mainly raises points of clarification in the technical exposition which must be carefully addressed in any final version.  They ask clarification questions about the experimental design that are well answered in the rebutall and the initial score to weakly accept is maintained.

TJux would like to see deeper theoretical discussion in the paper and has some questions around how the secret is embedded in the complex (FFT) domain. I think it is a reasonable criticism – this is not well explained in the paper, but ultimately this is because there is not a direct encoding or theoretical basis but simply a backprop optimization to perturb the complex numbers.  The reviewer seems satisfied with the response.

Overall I think this is a borderline acceptable paper.  The method is somewhat novel in its approach to directly perturb the latent signal in FFT space.  It is not a massive innovation but the experimental work is complete enough that it will provide some knowledge to the community of how far such an approach can be pushed to add robustness to a watermark.  Based on the above summary I consider all the criticisms were addressed and some important experimental results added, all of which must be present in the camera ready copy.  My recommendation is to accept the paper.